# Controlled open-cell two-dimensional liquid foam generation for micro- and nanoscale patterning of materials

Juyeol Bae [1], Kyunghun Lee [1], Sangjin Seo [1], Jun Gyu Park [1], Qitao Zhou[1] & Taesung Kim[1]

Liquid foam consists of liquid film networks. The films can be thinned to the nanoscale via evaporation and have potential in bottom-up material structuring applications. However, their use has been limited due to their dynamic fluidity, complex topological changes, and physical characteristics of the closed system. Here, we present a simple and versatile microfluidic approach for controlling two-dimensional liquid foam, designing not only evaporative microholes for directed drainage to generate desired film networks without topological changes for the first time, but also microposts to pin the generated films at set positions. Patterning materials in liquid is achievable using the thin films as nanoscale molds, which has additional potential through repeatable patterning on a substrate and combination with a lithographic technique. By enabling direct-writable multi-integrated patterning of various heterogeneous materials in two-dimensional or three-dimensional networked nanostructures, this technique provides novel means of nanofabrication superior to both lithographic and bottom-up state-of-the-art techniques.

[1] Department of Mechanical Engineering, Ulsan National Institute of Science and Technology (UNIST), 50 UNIST-gil, Ulsan 44919, Republic of Korea. Correspondence and requests for materials should be addressed to T.K. (email: tskim@unist.ac.kr)

Liquids continuously deform under shear stress[1]. This physical characteristic allows the liquid parts of material inks to be shaped freely with physicochemical forces, benefiting evaporative bottom-up assembly of target materials in inks into desired patterns; typically, this process is called liquid-mediated patterning (LP). Historically, early strategic tools for the liquid shaping, such as oriental brushes and printing presses have significantly affected civilization[2,3], and recent advancements in microfluidic and nanofluidic for spatiotemporal manipulation of microscale and nanoscale fluids[4] have substantially enriched studies on functional materials[5–8]. In particular, natural liquid foam systems observed in soap, beer, and even embryogenesis[9] can be advantageous compared to reported microfluidic and nanofluidic LP (MNLP) techniques in comprehensive perspectives of cost, dimensions, controllability, and scalability. It is typical that MNLP techniques for structuring nanowires utilize liquid–air interfaces without templates[10,11] or with micro-structured[12–14] or nanostructured templates[15–17], and representative examples of their use include flexible electronic devices, such as sensors, logic gates, and conductors[18]. However, the state-of-the-art methods require expensive preparation steps for templates[15]; otherwise, the spatial control of the liquid is low-resolution[11]. Microstructured templates, which provide a compromise between the cost and controllability issues[19], are also limited in that it is only possible to produce simple unidirectional one-dimensional (1D) structures[12].

To this end, the network of thin liquid film in two-dimensional (2D) foam is a powerful alternative for producing 2D or three-dimensional (3D) micro-/nanostructures in a well-ordered, large-area, and networked format[20–22]. However, several key challenges remain to be solved before liquid foam can be effectively utilized. First and most importantly, the topology of liquid foam spontaneously varies in a complex and restricted manner during evaporative change of the liquid volume fraction, due to combination of Plateau's law and Ostwald ripening[23,24]. Second, control of liquid morphologies with free surfaces remains difficult[19], because of the dynamic fluidity (e.g., capillary action and Marangoni effect) associated with interfacial forces[25] and the scaling law[26]. With regard to these two issues, Huang et al. recently suggested the use of a post array for modulating the direction of the Ostwald ripening, demonstrating a controllable and designable liquid film network[22]. Third, despite scientific breakthroughs in the control of 2D foam arrays, it appears that methods based on the natural foam are technologically limited for patterning applications from the viewpoints of process time, patternable materials, reliability, multi-patterning, and compatibility with common fabrication processes. Because natural foam is the closed system in which active physical access to manipulate foam is unavailable. As an exemplified weakness, the closed system is not free from very low effective evaporation flux, such that the patterning is very long and tedious processing time is proportional to the patterning area (typically 0.5–1 h for 1 cm$^2$).

To address these issues, we devised a micro-/nanofluidic liquid-mediated patterning (MNLP) technique to innovate 2D liquid foam from the conventional closed system (i.e., closed-cell 2D liquid foam) into an unprecedented open system (i.e., open-cell 2D liquid foams). We are motivated by the potential of the open system for patterning liquid-mediated materials at the microscale and nanoscale. Our approach utilizes a microhole array overlaid on a micropost array. The microhole array provides accessible paths to an atmospheric environment for the foam cells, which not only allows the spontaneous directed evaporative growth of discretized liquid–air interfaces but also completely excludes the Ostwald ripening. Instead, the micropost array anchors the grown interfaces at predefined locations in a controlled manner. To the best of our knowledge, we are

the first to investigate the mechanism of open-cell 2D liquid foam and characterize the MNLP technique to shape various liquid film networks. Through evaporative thinning of the network to confine liquid-mediated or aqueous materials at the microscale and nanoscale, we show that liquid-mediated materials can finally be assembled into permanent well-ordered micro-/nanostructure patterns. We further demonstrate that the MNLP technique can be not only repeatedly used, but also can be combined with the conventional UV-writing process, making it possible to fabricate multi-integrated, heterogeneous, and mixed-scale material patterns including 2D or 3D nanostructures on flexible substrates in a simple, low-cost, scalable, and mask-less manner.

## Results

**Principle of engineered 2D liquid foam.** Figure 1 shows the microfluidic device for engineering 2D natural liquid foam and its key working principles. The device is prepared by layering a film of polyurethane acrylate (PUA) through-hole membrane on a polydimethylsiloxane (PDMS) frame having a micropost array (Fig. 1a); the detailed fabrication process is summarized in Supplementary Fig. 1 and in the "Methods" section. The microposts support the flexible PUA membrane, forming a hexahedral space where liquid can be filled and evaporated through the microholes (Fig. 1b). The holes and posts are fabricated to be 60 μm in diameter and 25 μm in height with a spacing of ~100 μm. The micropost array in the frame is ~8 mm × 8 mm and is placed to locate posts at the centers of the microholes in the membrane.

To observe the generation and movement of liquid–air interfaces during liquid evaporation from top view (Supplementary Movie 1, red highlights in Fig. 1c), we filled the device with a liquid containing a surfactant. For all the experiments, 2.3 μL of 1 mg mL$^{-1}$ Pluronic F-127 aqueous solution was used to characterize the engineered 2D foam, while the aqueous solutions mixed with particles or polymers were used for the material patterning. Discretized circular liquid–air interfaces are initially formed underneath the individual holes, shown as black bold circles at $t = 120$ s. Subsequently, the cylindrical interfaces uniformly expand laterally in a concentric manner surrounding the holes, face each other, and deform into a thin liquid film having a lamellar structure and Plateau borders (refer to Supplementary Fig. 2)[27]. The Plateau borders are finally pinned to the microposts, forming a grid pattern ($t = 260$ s); one post anchors four mutually perpendicular films. Further evaporation thins the films further ($t = 290$ s). The surfactant prevents the thin liquid films from rupturing by balancing the Laplace pressure with the disjoining pressure, as illustrated and explained in Supplementary Fig. 2[28].

Indeed, comparison of the above results with the natural foam show that the micropost and microhole are critical in engineering the complex physical system of nature into well-ordered liquid patterns (Fig. 1c). The 2D natural foam is generated with a common design of a droplet microfluidic device[29]. First, the micropost array in the engineered 2D liquid foam pins the films between the nearby microposts for ease of manipulation in subsequent patterning applications. By contrast, in the natural 2D foam, films continuously undergo uncontrollable spatial movements. Second, the pinning of liquid films allows a grid pattern with fourfold junctions of Plateau borders, which is not available in natural 2D foam because of Plateau's law (only threefold junctions with mutual angles of 120° are formed to minimize surface energy)[23]. Third, microholes exclude the Ostwald ripening phenomenon (foam coarsening by the diffusion of gas across the interfaces from high to low Laplace pressures). Ostwald ripening causes uncontrollable topological changes called *T*1 and

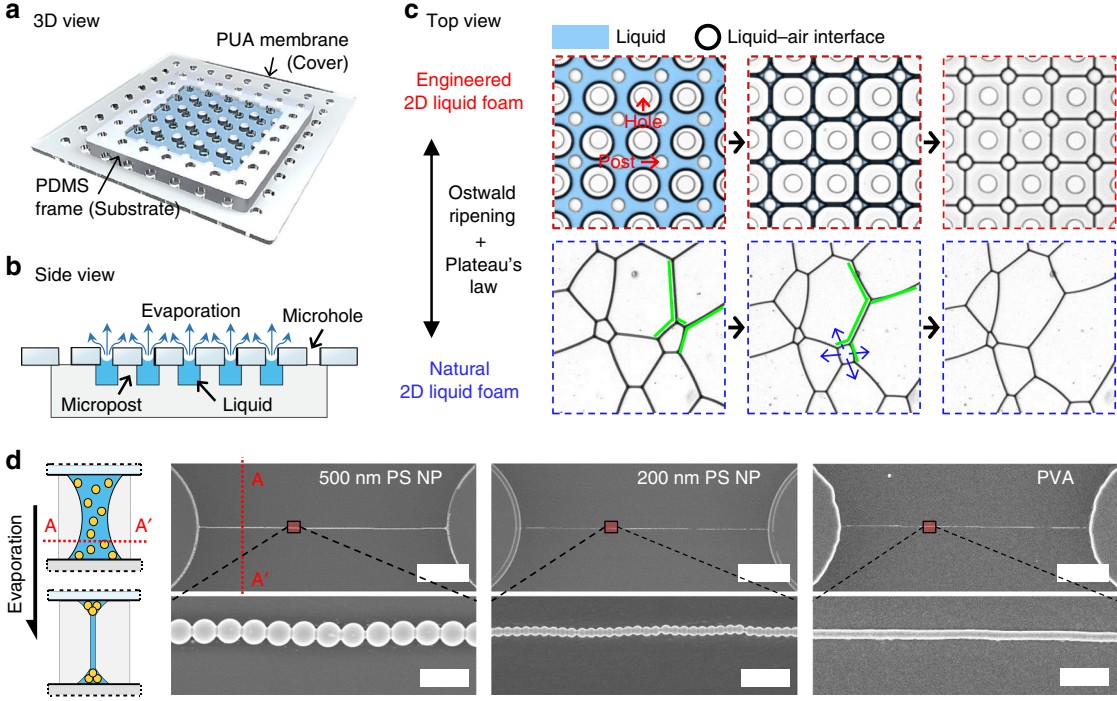

**Fig. 1** MNLP technique to engineer two-dimensional natural liquid foam for structuring materials. **a** Schematic of the microfluidic device in 3D view. The PDMS frame having microposts is covered with the through-hole PUA membrane. **b** Cross-sectional side view of the device. Liquid filled in the device (sky-blue) can be evaporated through the microholes. **c** Comparison between generation/dry processes of engineered 2D liquid foam (top, red) and natural 2D liquid foam (bottom, blue). Uniform pattern of liquid films in fourfold junctions are possible in the engineered form, being produced against Plateau's law and without Ostwald ripening. The blue region is the liquid. The black-bold lines are liquid–air interfaces. The white circles in the blue region and within the black-bold lines are microposts and membrane holes, respectively, as indicated with red arrows. In the natural foam, green liquid films are in the $T1$ transition, and blue arrows describe the $T2$ transition. **d** SEM images of structured solid materials, in consequence of assembly process during evaporation of the engineered foam. The schematic of A–A' cross-sectional side view depicts the principle of the MNLP process. The scale bars in top and those in bottom are 20 and 1 μm, respectively

$T2$ transitions in 2D natural liquid foam[30], coupled with Plateau's law. The $T1$ transition corresponds to the perpendicular rearrangement of two adjacent threefold junctions, as shown in green highlights in Fig. 1c. The $T2$ transition corresponds to the consumption of smaller air cells by larger adjacent air cells, as shown by the blue arrow in Fig. 1c. These topological changes do not occur in the engineered liquid foam because the air cells remain at atmospheric pressure through the hole array, yielding the exclusion of Ostwald ripening. Lastly, the processing time for the liquid pattern is short (<3 min) regardless of the pattern area size because of the effective evaporative outlets offered in 2D space.

To structure patterns of suspended particles or dissolved polymers, we used films of the engineered foam as molds. Figure 1d shows self-assembled polystyrene (PS) nanoparticles (NPs) and polyvinyl alcohol (PVA) on the PDMS frame (see also Supplementary Fig. 3). The materials are constrained within the liquid as evaporation of the engineered foam continues. Thereafter, at some point when all the liquid has dried, the particles or polymers remain, and are assembled into comparable structures with liquid morphology. PS NPs of 200 and 500 nm were patterned into grids of lines between nearby microposts. The line widths ranged from 10 μm to 200 nm, and their cross-sections were ridge-like. Notably, the PS NPs were well ordered and self-assembled even at single-particle resolution. The pattern structure of water-soluble polymers, such as PVA, polyacrylic acid (PAA), and dextran was similar to the PS NPs pattern, implying that the MNLP process can be applied to a wide range of materials. Simultaneously, the same material patterns were formed on the bottom side of the PUA membrane.

**Characterization of engineered 2D liquid foams**. We characterized the MNLP process in terms of the radius ($r_p$) and height ($h_p$) of the posts, distance between two neighboring posts ($d$), configuration of the post and hole arrays, and relative humidity (RH) ($\phi$) (Fig. 2). First, we found that $r_p$ and $d$ are critical in obtaining well-ordered liquid patterns. Figure 2a shows mutually contacting air cells in the device with post and hole arrays in a square format; the format is defined as 4P1H (four posts and one hole as the unit of configuration). The radius of the air cells ($r_a$) is determined by $r_p$ and $d$, i.e., $r_a = (r_p + d/2)\cdot\tan(\theta/2)$, where $\theta$ is the angle of the polygon configuring the post array (e.g., for 4P1H configuration, $\theta = \pi/2$). The ideal radius of the post (indicated with a red-dot circle) at the center of the confined liquid space (blue region) is $r_i$, and its geometrical relationship is $r_i/(d/2 + r_p) = \sec(\theta/2) - \tan(\theta/2)$. In the 4P1H configuration with $r_i < r_p$ ($r_p = 60$ μm, $d = 50$ μm, and $h_p = 25$ μm), a well-ordered grid pattern is formed (Fig. 2b). Meanwhile, when setting $r_i > r_p$ by varying only $d$ ($r_p = 60$ μm, $d = 200$ μm, and $h_p = 25$ μm), we can see threefold junctions of liquid films aside from microposts, i.e., defects. We analyze the defect generation according to changes in $r_p$ values at $d = 130$ or $200$ μm and $h_p = 25$ μm, counting the number of defects over the entire frame (Fig. 2c, Supplementary Fig. 4). The defects increase with the increase in the $\Delta r$ ($=r_i-r_p$) value. In addition, when $\Delta r$ is nondimensionalized by dividing by $r_a$, the two graphs are well superimposed, implying that $r_p$ should be greater than $r_i$ to pin the interfaces before they make mutual contact. Otherwise, the interfaces mutually push and deform asymmetrically until they are constrained by solid boundaries. In other words, the case of $r_i > r_p$ is analogous to the natural 2D liquid foam in the physically non-constrained space shown in

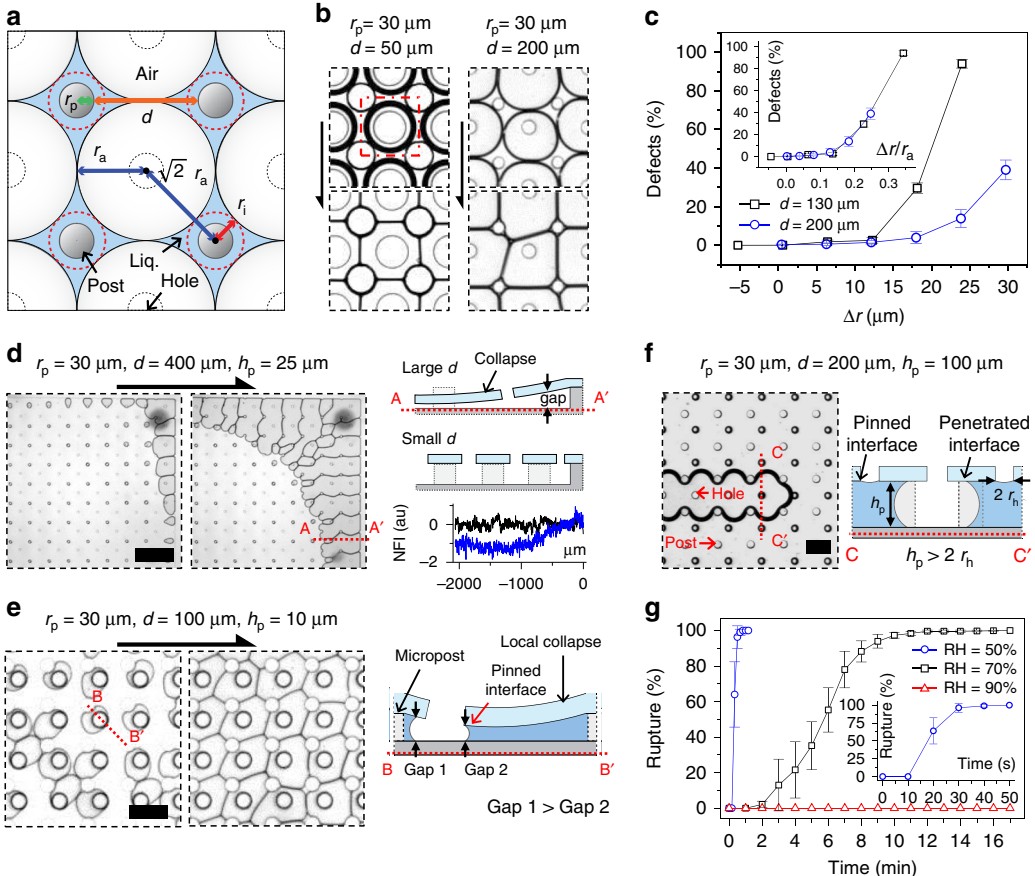

**Fig. 2** Characterization of geometrical and environmental parameters. **a** Modeling the generation of engineered liquid foam. **b** Generation of defect-free (left) and defective (right) patterns with respect to the $d$ value ($h_p = 25 \mu m$). **c** Quantification of defect generation over the entire space of micropost array with varying $r_p$ values ($d = 130$ or $200 \mu m$ and $h_p = 25 \mu m$). The inset graph shows a superposed relationship when $\Delta r$ is nondimensionalized. Error bars represent the standard deviation over the entire patterning area. **d** Undesired drainage toward center of the frame when $d$ is increased to $400 \mu m$ (left). Schematic (right and top) and graph (right and bottom graph) show the results caused by collapse of the membrane. The conceptual illustration is the cross-section A–A′ in the side view of the device. In the graph, the normalized fluorescence intensity (NFI) of the FITC solution in the device whose $d = 100 \mu m$ (black) and $400 \mu m$ (blue) is measured from the edge (set as origin) of the frame. The scale bar is $900 \mu m$. **e** Asymmetric expansion of the interfaces when $h_p$ is decreased to $10 \mu m$ (left), and the conceptual illustration depicting the side view of the cross-section B–B′ (right). The scale bar is 150 $\mu m$. **f** Expansion of the penetrated interfaces when $h_p$ becomes larger than the diameter of the holes (left), and conceptual illustration depicting a side view of the cross-section C–C′ (right). The scale bar is $200 \mu m$. **g** Quantification of liquid films under different humidity conditions. We counted number of ruptured liquid films out of the initial liquid films. Error bars represent the standard deviation over the entire patterning area. The inset graph shows a clearer view of the result when RH = 50%. Overall experimental parameters are summarized in Supplementary Table 1

Fig. 1c, until the interfaces become large enough to interact with the microposts. According to the same physical principle, when the 3P1H configuration is changed to $r_i > r_p$ from the condition of $r_i < r_p$, the well-ordered triangular liquid pattern changes to a chaotic liquid pattern (Supplementary Fig. 5).

We further characterized the geometrical conditions. When the $d$ value is larger than $400 \mu m$, the expansion of interfaces proceeds from the edge to the center of the PDMS frame in a non-uniform manner (Fig. 2d). This may be attributed to the decreased number of posts (see the illustration in Fig. 2d). A flexible membrane may collapse under surface tension when $d$ is large, resulting smaller gaps near the center than those near the edge of the frame. This is supported by the fluorescence intensity measurement. When the frame was filled with a fluorescein isothiocyanate (FITC) solution, the fluorescence intensities were not constant across the frame from the edge to the center (blue line in the graph). As the fluorescence intensity is proportional to the thickness of the solution, we can infer that the membrane may collapse when $d = 400 \mu m$; when $r_p = 30 \mu m$, $d = 100 \mu m$, and $h_p = 25 \mu m$, the intensity is constant (black line in the graph).

According to the Young–Laplace equation, the expansion of the liquid–air interfaces in the larger gap occurs in prior to interfaces in the smaller gap[31]. From these results, it can be inferred that other structural parameters, such as the radius and height of the post/hole and the mechanical properties of the membrane/frame can influence the collapse of the membrane and then the MNLP process.

We additionally investigated the effect of the $h_p$ value on the MNLP (Supplementary Fig. 6). When $h_p$ decreased from 25 to 10 $\mu m$ at $r_p = 30 \mu m$ and $d = 50$ or $100 \mu m$, the liquid–air interfaces expand not to form a perfectly circular shape but rather a distorted shape. Depending on the degree of distortion, either ordering (Supplementary Fig. 6a) or non-ordering (Fig. 2e) resulted. The distortion may be attributable to the increased capillary force acting on the surface of the membrane, which is in inverse proportion to the $h_p$ value, which is a well-known physical relationship[32]. The increased force induces the local collapse of the membrane between the nearby posts, thus causing the difference in the gaps at the edges of the relatively misaligned holes (see the illustration in Fig. 2e). If this difference exceeds a

certain threshold, the liquid–air interfaces in the larger gaps recede in prior to interfaces in the smaller gap, as mentioned above[31]. On the other hand, such local collapse did not cause influential difference to the gaps when $h_p = 25\,\mu m$ at $r_p = 30\,\mu m$ and $d = 50$, 100, or 200 $\mu m$. Additionally, because of the relation between $h_p$ and the membrane collapse, the same result shown in Fig. 2d occurred even at $d = 200\,\mu m$ when $r_p = 30\,\mu m$ and $h_p = 10\,\mu m$ (Supplementary Fig. 6c). Conversely, when $h_p$ increased from 25 to 100 $\mu m$ at $r_p = 30\,\mu m$ and $d = 200\,\mu m$, only the accidentally penetrated interface expands, while those interfaces at the bottom side of the holes are pinned (Fig. 2f). This is because the penetrated interface in the larger gap ($h_p$) recedes prior to those in the smaller gap (i.e., the diameter of the hole) in the same manner as mentioned above (see illustration in Fig. 2f)[31]. This explanation is supported by the fact that a well-ordered liquid pattern is produced when the diameter of the hole becomes larger than $h_p$ (Supplementary Fig. 6i). Therefore, we designed the device by considering the structural stability of the membrane and the priority rule for the receding interfaces. Unless otherwise stated, $r_p$, $r_h$, $d$, and $h_p$ are 30, 30, 100, and 25 $\mu m$, respectively, in the following experiments.

We tested the effect of RH on the MNLP process (Supplementary Move 1). After generating and maintaining liquid patterns at RH = 95%, we observed the response of the patterns at RH values of 50%, 70%, and 90% over time (Supplementary Fig. 7). When RH = 90%, the patterns become slightly thinner and last longer than 12 h; however, when RH = 70% and 50%, the liquid films rupture. We counted the number of ruptured drying films over time and plotted the results (Fig. 2g). The drying speed is inversely proportional to the RH value. Interestingly, when the RH varies from 90% to 95%, the liquid films at different thicknesses remain stable for infinite time.

**Design of various liquid patterns**. To design various kinds of liquid patterns, we investigated the effect of the post and/or hole array's configurations (Fig. 3, Supplementary Movie 2). We designed micropost arrays in square, hexagon, and triangular unit configurations (indicated using red dashed lines), in which microholes were placed at the centers of the unit polygons. The locations where the posts were removed are indicated using blue circles, whereas those where the holes were removed are indicated using green circles. For the 4P1H (Fig. 3a) and 6P1H (Fig. 3b) configurations, the patterns of the networked liquid films are the same as those of the unit polygons. In the same 3P1H configuration with one shown in Supplementary Fig. 5 ($r_i \gg r_p$), defects are observed (Fig. 3c). Moreover, we investigated the effect of the removal of certain posts on the pattern formation. When the center post in the 4P1H configuration is removed, the liquid pattern showed two interconnected threefold junctions of Plateau borders (Fig. 3d). On the other hand, when the two posts in the 6P1H configuration are removed in a similar manner, the generation of the honeycomb-like liquid pattern shown in Fig. 3b is unaffected (Fig. 3e). The two cases show different results because liquid interfaces without physical constraints must obey Plateau's law. In the 6P1H configuration, regardless of what posts are removed, the three liquid interfaces always mutually meet in accordance with the law, without any rearrangement of the interfaces. For this reason, even without any microposts, a honeycomb-like liquid pattern could be formed (Fig. 3j). In contrast, in the 4P1H configuration, the interfaces should deform asymmetrically and then be rearranged in accordance with Plateau's law. Therefore, the pinning posts are important in designing liquid patterns not restricted by Plateau's law.

Next, the size of the holes surrounding the post array and removal of the holes are also design factors for obtaining various patterns. When a larger and double-spaced hole array surrounding certain posts is employed in the micropost array of Fig. 3a, a double-sized square pattern is formed (Fig. 3g). In a similar manner, the same honeycomb liquid pattern shown in Fig. 3b is designed even on the micropost array of Fig. 3c (Fig. 3h). Thus, the posts surrounded by the holes do not contribute to the pattern formation, because the interfaces do not meet the pinning posts. In addition, when certain holes are periodically removed with consideration of the geometrical parameters and Plateau's law, interesting liquid patterns can be designed. For example, when the three holes (green circles in magenta-dashed lines in Fig. 3f, i) are periodically removed from the configuration in Fig. 3c, well-ordered hexagonal (Fig. 3f) or enlarged triangular patterns (Fig. 3i) are formed; the Y-shaped threefold junction of films is the basic component in these two cases. Interestingly, if we were to shape the holes into a word like 'NATURE,' we could shape the liquid–air interfaces of the air cells into that word (Fig. 3k).

**Potentials in materials patterning at microscale and nanoscale**. To integrate multiple patterns of heterogeneous materials on a single substrate, we perform iterative processes of the MNLP technique (Fig. 4a). First, one membrane is repetitively reused to pattern multi-stacked heterogeneous materials (Fig. 4b). For example, we patterned materials firstly with a 13 mg mL$^{-1}$ suspension of 500 nm PS NPs in the 4P1H configuration, and then we repeated the process using a 6.5 mg mL$^{-1}$ suspension of 200 nm PS NPs. The patterned materials were sintered at 90 °C for 2 min after each iteration of the patterning process, thus preventing their destruction from the viscous drag generated by convection flow during loading of additional liquid samples. This multiple MNLP process yields a core–shell line pattern; the former NPs are self-assembled in a ridge-like shape and then covered with the latter NPs. Second, different membranes are used in series on a single PDMS frame to integrate different patterns of heterogeneous materials (Fig. 4c, Supplementary Fig. 8). For example, we firstly patterned a 1 mg mL$^{-1}$ suspension of 200 nm red fluorescence (RF) PS NPs in the same configuration with Fig. 3h, and then we patterned a 1.5 mg mL$^{-1}$ suspension of 380 nm green fluorescence (GF) PS NPs in the same configuration with Fig. 3f. As a result, a red honeycomb pattern was well-superposed with a green Y pattern (Fig. 4d). The merged fluorescence images and scanning electron microscope (SEM) images confirm that no influential cross-contamination occurs between the two materials during the multiple MNLP. Each individual pattern clearly comprises a single material.

To further demonstrate the potentiality of the multiple MNLP, we constructed two more integrated patterns that are possibly meaningful in building functional devices. A periodic triangular pattern comprising three heterogeneous PS NPs was formed (Supplementary Fig. 9a); we firstly patterned 0.65 mg mL$^{-1}$ suspension of 100 nm GF PS NPs in the same configuration with Fig. 3h; then, we changed the position of the used membrane and again patterned a 0.9 mg mL$^{-1}$ suspension of 200-nm RF PS NPs; finally, a 1.25 mg mL$^{-1}$ suspension of 500 nm blue fluorescence (BF) PS NPs were patterned in the same manner. The resultant pattern is possibly comparable to patterns extensively used in red-green-blue quantum dot light-emitting diode applications[33]. Perpendicularly crossed junctions of two heterogeneous PS NPs were formed (Supplementary Fig. 9b); 0.675 mg mL$^{-1}$ suspension of 200-nm RF PS NPs and 0.625 mg mL$^{-1}$ suspension of 500 nm BF PS NPs were sequentially patterned in a similar manner as described above. The resultant pattern is possibly comparable to the structure of nanoelectronic devices[34]. Each material is very precisely self-aligned as guided by the membrane and post array. These demonstrations are partially

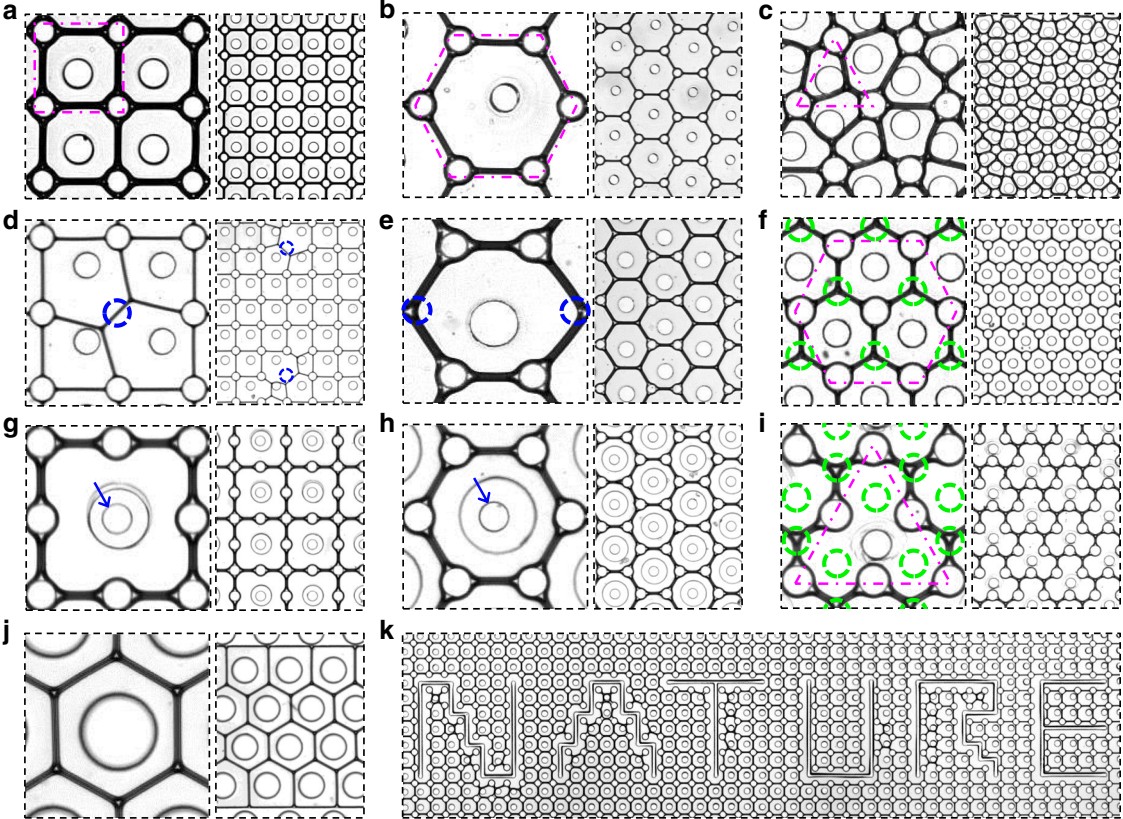

**Fig. 3** Various liquid patterns designed by configuring the arrays of microposts and holes. **a** Liquid pattern in the 4P1H configuration. **b** Liquid pattern in the 6P1H configuration. **c** Liquid pattern in the 3P1H configuration. **d** Modification of the 4P1H configuration; the microposts (indicated using blue circles) in the 2 × 2 units of the square are removed. **e** Modification of the 6P1H configuration; the two microposts (indicated using blue circles) in the unit hexagon are removed. **f** Modification of the 3P1H configuration; the three microholes (indicated using green circles) in the unit space (indicated using a magenta line) are removed. **g** Modification of the microhole array of the 4P1H configuration; larger and double-spaced microholes are employed individually at the centers of 2 × 2 units of the square. The blue arrow indicates a micropost surrounded by a double-spaced microhole. **h** Modification of the microhole array in the 3P1H configuration; larger and double-spaced microholes are employed individually at the centers of unit hexagons. **i** Modification of the 3P1H configuration; the three microholes (indicated using green circles) in the unit space (indicated using a red line) are removed. **j** Modification of the 6P1H configuration; a post array is removed while a 100-μm-thickness membrane was overlaid on 1-mm channels to prevent the membrane collapse. **k** The 4P1H configuration is modified with some holes shaped into the word 'NATURE.' Overall experimental parameters are summarized in Supplementary Table 1

competitive, compared with widely used stamping-based patterning which requires cumbersome high-resolution alignment[33]. Hence, the MNLP's capability to integrate multiple patterns of organic or inorganic functional building blocks into any substrate, such as a flexible PET film, is expected to advance the development of transparent or flexible multi-functional electronic devices.

To selectively solidify the produced liquid pattern, we combined a lithographic technique with the MNLP technique. We used 0.01 mL of poly(ethylene glycol) diacrylate (PEGDA) in 1.5 mL of the Pluronic F-127 solution containing a photoinitiator. The solution is patterned in the 4P1H or 3P1H configuration and is maintained without film rupture at RH = 90% in a nitrogen environment. Then, ultraviolet (UV) light (using a ×10 objective lens) through an adjusted aperture is illuminated onto the prepared liquid pattern in a similar manner with the direct-writing technique (Fig. 5a)[35]. As a result, we selectively photo-crosslink the PEGDA solution along a letter "J" (Fig. 5b). The width of the UV-written pattern can vary from a single film to several films by adjusting the size of the aperture and magnification of the lens (Fig. 5c). With a programmed motorized microscope stage, the word "UNIST" is produced at single-film resolution in an automated manner (Fig. 5d). Therefore, we can infer that liquid-mediated photo-curable

material inks are important in developing a simple, cost-effective, on-demand, fast, mask-less patterning process for nanowire network[36], thus providing an alternative to electron-beam lithography in perspective of cost and throughput. For instance, after the MNLP of photo-curable ink is done to pattern a nanowire structure in a grid format, UV-writing of the desired nanowire network can be performed even with micrometer-resolution focused UV-light.

Furthermore, the MNLP selectively allows patterns of 2D or 3D networked wires, depending on the adhesion between the membrane and the patterned materials (Fig. 5e, Supplementary Fig. 10a–c). In Case-I, the adhesion is lower than the mechanical strength of the cured PEGDA, in which the covered membrane was detached from the frame without destroying the pattern of PEGDA structure. Interestingly, no solidified PEGDA wall remained in the space where the lamella of the liquid film had existed. That is, the MNLP technique is a novel method for fabricating networked 3D patterns of suspended nanowires. Conventionally, such 3D structures were generally obtained using relatively laborious, multi-step, lithographic processes[37]. In Case-II, we partially cured the bottom surface of the PUA membrane to co-crosslink the PUA with the PEGDA during UV-writing. The co-crosslinking resulted in a tight bonding between them. The top parts of the

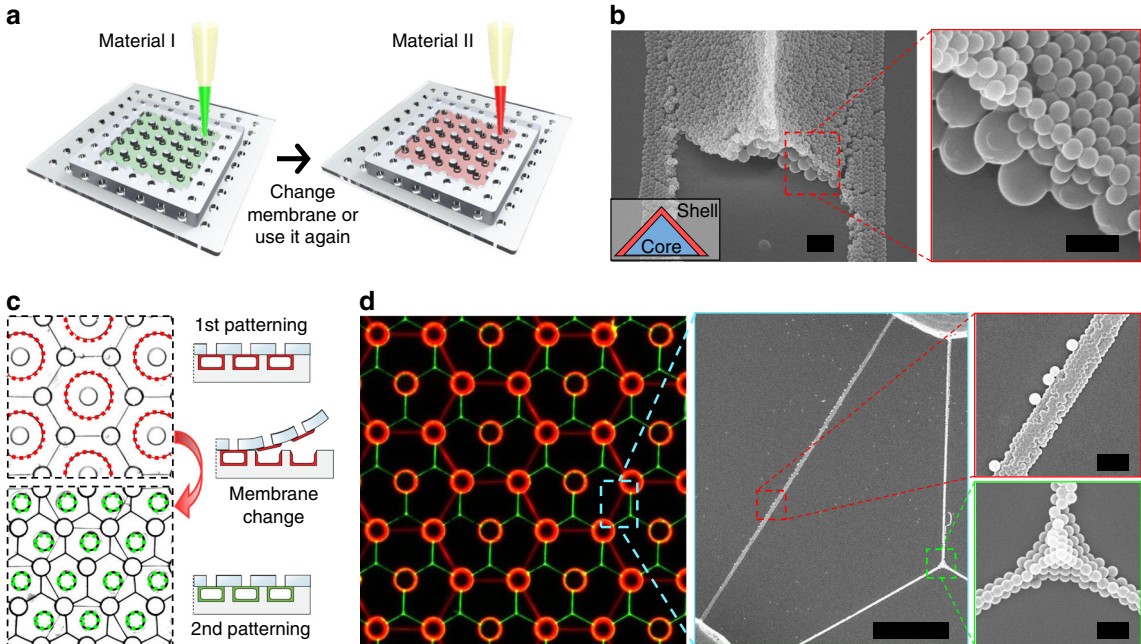

**Fig. 4** Multiple MNLP on a substrate for integrated patterns of heterogeneous materials. **a** Schematic of the multiple MNLP process. After material I is patterned on the frame, the used membrane can be changed or reused according to the desired pattern, as shown in **b**, **d**. **b** SEM images after the multiple MNLP when the membrane is reused after the first patterning. SEM image shows a core–shell triangular structure (core: 500 nm PS NPs; shell: 200 nm PS NPs) in oblique view (left) and enlarged view (right); the initially perfect linear structure is intentionally broken for imaging the core. The scale bars are 2 μm and 500 nm. **c** Bright-field images after the multiple MNLP. After the first patterning, the membrane (red-highlight hole array) is replaced with another type of membrane (green-highlight hole array) for second patterning. 200-nm RF PS NPs are structured into lines of honeycomb patterns, followed by 380-nm GF PS NPs pattern of lines forming threefold junctions. **d** Merged fluorescence and SEM images showing integrated patterns of heterogenous materials in **c**. The scale bars are 20 μm (left) and 1 μm (right two)

3D structures were torn out when the PUA membrane was peeled off. As a result, 2D-structured patterns were left on both the membrane and the frame. Of course, the UV-curing can be applied to the entire substrate to cure all the liquid films (Supplementary Fig. 10d). In addition, the widths of the PEGDA structures can be controlled by adjusting the mixing ratio of PEGDA to water (Supplementary Fig. 10e); regardless of the ratio, the thicknesses of the ridge-like structure's apex were found to be 200 nm.

## Discussion

We develop a microfluidic device to engineer a natural 2D liquid foam from a physical systematic point of view, whereas previous studies have attempted to manipulate natural liquid foams based only on a conventional closed system. In our unconventional open foam system, the pressure of all the air cells is equal to atmospheric pressure due to our use of a unique overlaid through-hole membrane. This condition allows the simplification of the foam physics by completely eliminating the Ostwald ripening, and revealing a new aspect of foam generation and control. A set number of discrete air cells in designated positions expand uniformly into the desired well-ordered 2D liquid foams on flexible substrates over a large area and in a scalable manner. Even new types of controlled film network (e.g., periodic three-fold junctions of liquid films such as those shown in Fig. 3e, f, i, j) are possible, which is difficult to be reproduced using closed-cell-based methods due to the inevitable inclusion of Ostwald ripening. It is noteworthy that open-cell structures have been the subject of only solid-foam studies not liquid-foam to date. In short, our open-cell 2D liquid foam is based on a physical system that is very distinguishing from a conventional closed-cell 2D liquid foam.

The generated 2D liquid films can be ultimately applied to the patterning of various materials into 2D or 3D networked microstructure and nanostructure, after being pinned and thinned at set constant positions by the micropost array. In fact, Huang et al. pioneered the control of foam beyond the Ostwald ripening[22]. However, we introduced different microstructures, such as a overlaid hole array and a contrasting, scientifically unprecedented physical system (i.e., open-cell 2D liquid foam), thus better satisfying the technological requirements of material processing than their method. First, the open-cell liquid foam is generated and controlled by pure spontaneous evaporation through the microholes. This physical approach allows a wide choice of patternable materials, 3-min short process regardless of the size of patterning area, and deterministic defect suppression. In contrast, the closed-cell foam was produced by gas-generating chemical reactions; the number of patternable materials is greatly restricted by the latent chemical degradation of materials caused by reducing agents or by-products; it requires long time to await the equilibrium state of the chemical reaction in addition to the slow evaporation rate of the closed system; the air cells grow in a non-uniform randomized manner, resulting in stochastic defects. Second, the overlaid membrane enables the multi-way control of the foam design for producing various patterns on a single post design, which is a more advanced technique than the one-way control based only on the micropost array. The iterative MNLP processes can precisely integrate multiple self-arranged patterns of heterogeneous materials, anticipating the potential to replace conventional material fabrication techniques requiring cumbersome high-resolution alignments and numerous processing steps. Third, we extend the potential role of the foam in LP fields, demonstrating a synergistic combination of the bottom-up MNLP with standard top-down techniques such as direct writing. We can thus envision nanofabrication technology superior to

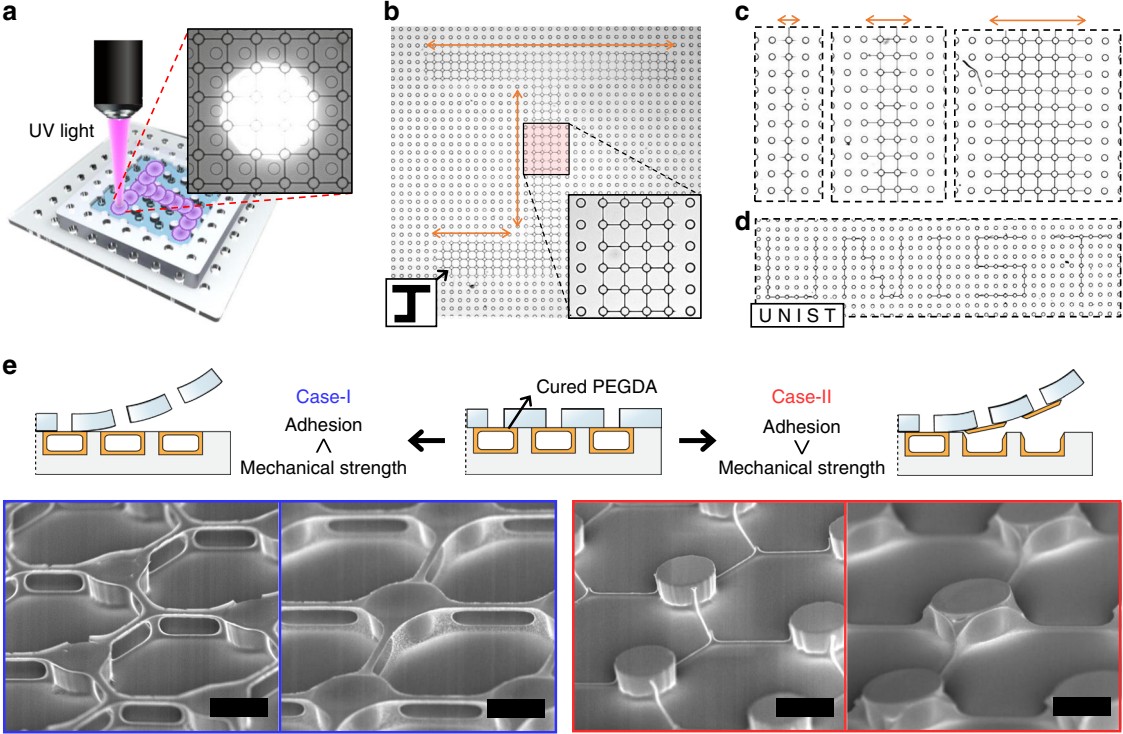

**Fig. 5** Combination of top-down direct-writing approach with MNLP. **a** Schematic of UV-writing process for selective solidification of patterned liquid films. The inset image shows spot of UV light illumination (bright circular region) on prepatterned grid of PEGDA solution films. **b** UV-writing of letter "J" on grid pattern of the PEGDA solution. The inset shows the enlarged image of solidified lines in UV-written letter "J." The orange arrows clearly identify the UV-written letter. **c** UV-written lines of PEGDA with respect to the adjusted size of aperture. **d** Demonstration of writing the letters in the word "UNIST" with single-film resolution. **e** Structuring of the UV-cured PEGDA into 2D or 3D in a selective manner. For Case-I (highlights with blue), the mechanical strength of the cured PEGDA is greater than the adhesion between the PEGDA and the membrane, resulting in a 3D structure with suspended wires. For Case-II (highlights with red), the mechanical strength is lower than the adhesion, resulting in a 2D structure without suspended wires. The scale bars are 50 μm

e-beam lithography in terms of both cost and throughput. Not unexpectedly, the engineered 2D liquid foam arrays can possibly be used as experimental tools for fundamental studies of the physicochemical properties of various liquid films at the microscale and nanoscale, such as the electrokinetics of liquid foam and transport characteristics of biological bilayer membranes. Therefore, the MNLP technique will help advance multidisciplinary research fields significantly by providing a visible practicality for studying the physics of foams and precisely patterning various materials at the microscale and nanoscale.

## Methods

**Reagents and materials**. The chemicals used in this work were obtained from Sigma-Aldrich, unless otherwise stated. In all the MNLP experiments, 2.3 μL of 1 mg mL$^{-1}$ Pluronic F-127 (P2443) aqueous solutions were used. For the evaporative deposition of the materials, the aqueous solutions were prepared by suspending or dissolving particles or solutes in the Pluronic F-127 solution. Fluorescent PS NPs with diameters of 200 nm (red, R200) and 380 nm (green, G400) were purchased from Thermo Scientific. Fluorescent carboxylated PS NPs with diameters of 100 nm (green, 16662), 200 nm (red, 19391), and 500 nm (blue, 18339) were purchased from Polysciences, Inc. PVA ($M_w$ = 9000–10,000, 360627), dextran ($M_w$~5220, 00269), and PAA ($M_w$~1800, 323667) were used. For the UV-writing, PEGDA ($M_n$~700, 455008) was dissolved in the Pluronic F-127 solution containing 2 mg mL$^{-1}$ 2-Hydroxy-4′-(2-hydroxyethoxy)-2-methylpropiophenone (410896) as the photoinitiator. The materials were processed at 25 °C in the MNLP. To quantify the difference in the gap between the membrane and the frame, 100 μM FITC (3326-32-7) in 1 × phosphate buffered saline (PBS, P5493) was used.

**Fabrication of the MNLP platform**. To fabricate the PUA membrane, we prepared a microfluidic mold (Supplementary Fig. 1)[38]. The PDMS part of the mold was designed to have microposts on which the microholes of the membrane would be formed. The PDMS part was prepared using a common soft-lithography process[39]

with a mixture containing a PDMS curing agent and a base (Sylgard 184, Dow Corning) at a ratio of 1:10. The prepared PDMS part was bonded to the glass by surface treatment with oxygen plasma. The microfluidic mold was filled with PUA (Young's modulus: ~300 MPa, MINS-311RM, Minuta Technology)[40] and then cured under UV exposure in a nitrogen environment, after which the microfluidic mold was subjected to vacuum at ~20 Pa for 20 min to prevent oxygen inhibition during the UV curing of PUA. To remove the cured PUA membrane from the mold, we cut a region of the plasma-activated bonding using a scalpel except for the post array and removed the post array by pulling the PDMS. Thickness of the membrane was 25 μm, unless otherwise stated. In parallel, a PDMS frame was also prepared using soft lithography. The frame was alternatively made with an optical adhesive (NOA 63, Norland Products). The surface of the PDMS frame was grafted with methoxy polyethylene silane (PLS-2011, Creative PEGWorks). The surface of the NOA 63 frame was grafted with [3-(trimethoxysilyl)propyl]methacrylate (440159, Sigma-Aldrich) in the UV-writing experiment. In other instances, the frames were simply treated with oxygen plasma. Finally, using an optical microscope, we manually slid the PUA membrane onto the PDMS/NOA frame using water as the lubricating layer, achieving alignment into the desired post-and-hole array configurations. After the perfect evaporation of the lubricant, the overlaid membrane conformally contacted the frame as a result of the van der Waals force interaction.

**Experimental setup**. The optical and fluorescent images were captured using a CCD camera (ORCA R2, Hamamatsu Photonics) mounted on an inverted fluorescence microscope (TI-U, Nikon; equipped with Intensilight C-HGFIE as the UV light source). The microscope was automated using a motorized stage (96S209-N2), a motorized focus controller (99A400), and a controller system (MAC 5000), which were manufactured by Ludl Electronic Products. The SEM images were obtained using a field-effect scanning electron microscope (FE-SEM, S-4800, Hitachi). To control the atmospheric conditions, a customized humidity/temperature-controlling system, comprising solenoid valves (S10MM-20-24-2, Pneumadyne Inc.), a humidity/temperature sensor (SHT15, SENSIRION), a microcontroller board (Arduino Uno, Arduino cc.), and a microscope incubator (CU-501, Live Cell Instrument), was programmed using LabVIEW software (National Instruments).

## Data availability

All data supporting this study and its findings are available within the article and its Supplementary Information, or they are available from the corresponding author upon reasonable request.

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

## Acknowledgements

This work was supported by a grant from the National Research Foundation of Korea (NRF) funded by the Korean government (MSIT) (NRF-2017R1A4A1015564 and NRF-2017R1A2A1A17069723).

## Author contributions

J.B. and T.K. conceived the idea and drafted the manuscript. Under the supervision of T. K., J.B. designed all the experiments and the numerical simulation. J.B., S.S., J.G.P., and Q.Z. performed the experiments. K.L. conducted the numerical simulation. All the authors discussed the results and approved the manuscript.

## Additional information

**Competing interests:** The authors declare no competing interests.

