## [Peer Review File · Nature Communications]

Reviewers' comments:

Reviewer #1 (Remarks to the Author):

The manuscript reports on micro/nanofluidic approach for fabrication of 2D liquid foam arrays, which has a potential in patterning the materials in liquid at the micro- and nanoscales. In addition, the mechanism of 2D liquid foam generation with exclusion of Ostwald ripening was discussed. The manuscript is clearly written and the conclusions are adequately supported by the obtained data. However, the manuscript is not suitable for publication in Nature Communications because of lack of novelty. There is a report in the same journal ("A general patterning approach by manipulating the evolution of two-dimensional liquid foams", Z. Huang et al., Nature Commun., 2017, DOI: 10.1038/ncomms14110), in which a strategy to manipulate the evolution of 2D liquid foams beyond Ostwald ripening is described and closely related results are obtained. Unfortunately, the authors did not refer to this publication.

I expect that the manuscript can be suitable for publication in journals in the field of colloids and interfaces, e.g. Colloids and Surfaces A, J. Colloid and Interface Sci., Langmuir.

Reviewer #2 (Remarks to the Author):

I enjoyed reading this manuscript. The authors used clever microfluidic designs to stabilize a monolayer bubble emulsion (foam) during evaporation : air vents to allow vapor escape, pillars to stabilize against Ostwald ripening and geometric optimization to pin the lamellae. The result is a beautiful hexagonal lamella structure without topological defects, whose films are sub-micron in dimension. With colloids within the film, patterned colloidal aggregations result after evaporation, with micro-lattices and nanoscale sheets.

I believe the paper is suitable for publication in Nature Communication, if these designs are new. I am not sufficiently familiar with the literature to assess if this is true. My only technical criticism is on the arguments of how smaller channel height can pin the meniscus by ensuring the bottom contact angle does not exceed the advancing contact angle. The specific advancing contact angle for their surface was not reported and it is not clear if it even exists for the hydrophilic substrate used. It is also not clear that the simple interfacial shape depicted is actually observed. Surely when the lamellae and plateau are formed, such a simplistic interface model is not accurate and the advancing contact angle argument breaks down, as the contact line is surely receding. This argument can be tightened more. A better argument is that the expanding bubble will make contact with the bottom substrate before the interface at the pinned location makes contact with the top substrate. This ensures that the meniscus curvature will always render the liquid pressure negative due to capillarity, thus suppressing any liquid pressure gradient that can prematurely drive meniscus motion.

Reviewer #3 (Remarks to the Author):

The manuscript relates to a method of making complex micro/nanoscale patterns using controlled drying of 2D foams trapped between a structured microtextured surface, and a microporous surface. The alignment of micropores on the microposts allows for nearly uniform evaporation of the foam resulting in naturally thinned nano/micro-liquid patterns pinned along the microposts. Such liquid patterns can be used to make solid patterns by simply adding either nanoparticles or a water-soluble polymer into the evaporating solution. The authors have demonstrated that by cleverly playing between the geometrical features of the underlying micropost surface and the overlaid microporous

surface, multi-patterned nano/micro-structures can be obtained. The manuscript is quite well-written, the authors have done quite exhausted study of different parameters that can influence the design of nano/micro-structures with the technique. To the reviewer, the manuscript appears to be suitable for the journal - provided the authors address the concerns mentioned below.

- To be clear, the idea of using foam evaporation guided by the micropost surfaces is not new. It has previously been demonstrated in the manuscript titled "A general patterning approach by manipulating the evolution of two-dimensional liquid foams". It is unclear why the authors did not cite this work. This is unacceptable. Although the basic idea has been demonstrated before, the reviewer does appreciate the use of overlaid microporous surface to control the foam evaporation and how that provides as an extra-control over the design of the nano/micro-structures one can obtain using this method.
- The authors appear to have done quite some extensive experiments varying the post radius, distance, hole radius etc. Depending upon such geometric parameters, different results are obtained. It would be very useful to the readers if the authors could provide a catalog of their experiments and the final image in the supplementary file.
- The authors should provide the images of the foam structures based on which Fig2c and Fig 2g were obtained. Such images could be part of the suggested catalog as mentioned above – or these could be separate figures in the SI itself.
- It is unclear whether Fig2c data was obtained by using samples with different r_p ? Authors should clarify that they sequentially varied the post diameter to obtain Fig 2c. The authors should also mention geometrical parameters (including h_p) of samples shown in the figure captions.
- Authors need to provide more description of the alignment of the PUA over PDMS structures. Authors describe PDMS/NOA frames – but it is unclear how these frames were used? Were they attached to PUA membranes? The thickness of PUA membranes is really thin. How did the authors avoid the folding of the structures and overlaying it over the structures?
- Some of the materials information is essential and should be included in the main text itself. For example, the solution used, the molecular weight of PDMS, PUA etc are crucial information. Such information can be included in the main text or the figure captions.
- Minor point: Instead of saying "under surface tension when d is long", it is better to say "under surface tension when d is large". Similarly, instead of using word 'Short' – use 'small'.
- Fig2d: The collapse of the overlaid membrane should also depend upon r_p and h_p . If r_p is large, then the membrane may not collapse. The reviewer imagines that there should be some d/r_p above which the collapse may occur. Can the authors provide a criterion on when the films would collapse as a function of geometrical parameters and the bending modulus of the overhang film?
- Fig 2e: The effect of h_p has been shown for a single experiment. Is it universal behavior? For all samples with $d < 400 \mu\text{m}$, do you obtain distorted foams for $h_p < 10 \mu\text{m}$? If not, the authors should clarify/correct/edit the statement.
- What happens if h_p is very large? Can authors provide some clarification on its possible effect?
- Minor: The authors state "were sintered at $90 \text{ }^\circ\text{C}$ for 2 mins". Was sintering done after each experiment?

- The reviewer is very curious to know why the Fig.4 experiments worked. Why did the first deposition not alter the evaporative dynamics of foam. In other words, it appears that the patterns made no difference in how the foams evaporated. Why? Was it because they are too small in size? Was there no pinning on these patterns?

Common Response Material to All Reviewers for the Novelty and Originality of This Work

In the revised manuscript, we emphasized many scientific and technological advances and new findings of our work compared to other relevant references reported by Huang et al. (Nat. Commun. 2017) and Tokuno et al. (Langmuir 2012). Here, we explain the principal and distinguishing features of our study in detail as follows, which are also summarized and compared in Table C1. We believe that these arguments will be helpful for all the reviewers to understand the novelty and originality of this work.

(1) Complete exclusion of Ostwald ripening (open-cell liquid foam) vs. modulation of Ostwald ripening (closed-cell liquid foam):

We developed a microfluidic device to engineer natural liquid foam from the viewpoint of a physical system, whereas reported studies have attempted to manipulate the natural liquid foam by using a conventional closed system. (Fig. R1). For a closed system (closed-cell 2D liquid foam), generated air cells have size-dependent pressure differences each other, because they are enclosed by rigid and gas-tight plates. It is inevitable for this physical condition to cause Ostwald ripening, but Huang et al. cleverly used the gaps/interval between nearby microposts to modulate the direction of Ostwald ripening. The interval of posts set the curvature of growing air cells to be small, when the liquid-air interfaces of the growing air cells encounter the posts. This allows abnormal growing of small air cells whose curvature is larger than the micropost-contacted large air cells (see Fig. 2a, b in their paper). For our open system (open-cell 2D liquid foam), the pressure of all air cells equilibrates to an atmospheric condition thanks to our unique use of an overlaid through-hole membrane. This physical condition allows complete elimination of Ostwald ripening, thereby suggesting a scientifically novel concept of foam generation. A controlled number of positioned discrete air cells expand uniformly into desired various liquid patterns. Then the pattern is pinned by a micropost array. To best of our knowledge, the open-cell 2D liquid foam should be a new system in foam physics. The open system will be interesting to interface research communities because such open-cell foam has been a scientific subject of only solid foam not liquid one. Moreover, the following arguments (2) ~ (5) show technological advances of our open-cell 2D liquid foam control.

Fig. C1. Physical comparison between two different systems of liquid foams. (a) For the closed cell, the

pressure difference between air cells induces the diffusion of gas, resulting in an Ostwald ripening phenomenon. (b) For the open cell, all the air cells have the same atmospheric pressure. This physical condition completely eliminates Ostwald ripening, resulting in no topological changes. Scale bars are 100 μm (left) and 200 μm (right), respectively. Some images are reproduced from the manuscript (Fig 1c, Fig. 3j).

(2) **Novel generation and control mechanism of uniform air cell arrays (physical evaporation vs. chemical reaction):** For the closed-cell liquid foam, a random number of non-uniform air cells in size are produced by a gas-generating chemical reaction like the decomposition of urea peroxide. This is a big hurdle in liquid-mediated patterning. Weaknesses are as follows: 1) Patternable liquid-dispersible materials are limited because they should get involved in the chemical reaction. Although hydrolysis of borohydride to make hydrogen gas is conceived as an alternative, the borohydride is a reducing agent which potentially can deteriorate the functionality of patterned materials. Of course, chemical by-products are also problematic. 2) It requires long time to await an equilibrium state of chemical reaction. Retardation of liquid evaporation should be accompanied with the reaction to maintain a stable equilibrium state, further increasing liquid-pattern processing time. This is evidenced by the sentences in the method section of the reference (“the system needed time to reach an equilibrium of gas concentration”; “To maintain bubble patterns for a long time, ~”) Moreover, the evaporation of the produced enclosed foam might take several hours for a 2 cm^2 area as reported by Tokuno et al., due to the lack of effective evaporative outlets. 3) The irregular air cell generation results in stochastic defects. No matter how small they make the interval of posts, there is at least a small probability of generating air cells smaller than the curvature of micropost-contacted air cells. The fact may cause proportional relation between the number of stochastic defects and the patterning area. Unfortunately, it seems that the study did not demonstrate and discuss large-area patterning at millimetre scales, although large-scale fabrication is an important factor in their transparent electrode applications. In contrast, our technique is based on the simple spontaneous evaporation through effective evaporative microholes, which makes it possible to not only generate uniform air cells but also control their number and position. In addition, our technique enables patterning of materials on flexible substrates, which will be more and more promising due to the rapid advances of soft electronics, and allows fast large-area, non-defective patterning of various solution-mediated functional materials such as biological samples, quantum dots, nanomaterials, photoresist, and so on.

- (3) **Various shapes of liquid film networks as designed including periodic threefold junctions:** In our technique, air cells can be patterned into not only any patterns as designed like a word ‘NATURE’ (Fig. 3k) but also periodic threefold junctions of liquid films (Fig. 3e, f, i, j). Notably, controlling the periodic threefold junctions are scientifically new, and it cannot be simply demonstrated by closed-cell liquid foam in which Ostwald ripening is inevitably involved. Although Huang et al. pioneered the way to producing various liquid patterns, it seems that they may not easily demonstrate such junctions. Moreover, we newly added data (Fig. 3j) that further supports the unique novelty of our technique that a liquid film network can be well controlled without any assistance of posts. Both the natural closed-cell liquid foam and the artificial liquid foam engineered by Huang et al. cannot produce and control this type of liquid patterns, while we produced these results in a simple manner.

Fig. C2. Various patterns having periodic threefold junctions of liquid films. Images are reproduced from the manuscript (Fig. 3f, i, j).

- (4) **Repeatable multiple integration and superposition of heterogeneous material patterns (multi-way control vs. one-way control):** It is obvious that the network of thin liquid films in 2D foam can provide an extremely low-cost alternative for producing the 2D micro-/nanostructures of functional materials in a well-ordered, large-area, and networked format. However, Huang et al.’s method can produce only one material pattern using a post design, because their passive post design allows one-way control of liquid foam. Here, overcoming their limitation, our method using overlaid through-hole membrane enables multi-way control of liquid foam on a post design for producing various patterns from a single post design. By repeating the patterning process on a single post design with various membrane designs, we precisely integrated self-aligned networks of multiple heterogeneous material patterns with a minimum number of processing steps. Multi-stacked heterogeneous materials were achieved as shown in Fig 4b. And, complex, multiple, and heterogeneous material patterns were superposed in a two- or more pattern format as shown in Fig. 4d and Supplementary Fig. 9. These results must be very interesting to the micro- and nanopatterning research community because conventional nanofabrication techniques require expensive and cumbersome high-resolution

alignments and even more complex processing steps to produce the same or similar structures.

Fig. C3. Example of integrated and self-aligned networks of multiple heterogenous materials. The unit triangle with three different materials (highlighted with coloured dot-lines) are periodically patterned. Images are reproduced from the manuscript (Supplementary Fig. 9).

- (5) **Versatile and synergistic compatibility of our technique with conventional micro- and nanofabrication techniques:** Our technique can be combined with a conventional microfabrication process such as a top-down direct-writing process, making it possible to write patterns like “UNIST” in 2D or 3D structure formats at micro- and nanoscales (Fig 5). Such a combination of the conventional top-down lithography and our bottom-up liquid-mediated patterning process can provide an unconventional, low-cost, high-throughput, and large-area nanofabrication technique, which is partly much better than standard e-beam lithography. Therefore, our technique shows remarkable potential for fabricating nanodevices such as nanoelectronics, optoelectronics, and nanogap-based sensors. [Redacted]

[Redacted]

Table C1. Comparison with other relevant literatures shows the novelty and difference of this work.

List	Tokuno et al., (Langmuir, 2012)	Huang et al., (Nat. Commun., 2017)	This work (2019)
Physical condition	Closed-cell liquid foam	Closed-cell liquid foam	Open-cell liquid foam
Patterning process	1) Vortex solution 2) Cover bubble with glass 3) Evaporation	1) Drop reactive chemical solution 2) Evaporation 3) Drop target solution 4) Cover with substrate 5) Evaporation	1) Assemble the frame and membrane 2) Capillary injection of solution 3) Evaporation
Control of Ostwald ripening	None	Post array - setting curvature of air cells - reversing direction of Ostwald ripening	Hole array - equilibrating pressure of air cells - eliminating Ostwald ripening completely
Control of fluidity	None	Post array (Liquid pinning)	Post array (Liquid pinning)
Air cell generation mechanism	Vortex mixing - uncontrollable - random size and number	Gas-generative chemical reaction - partially controllable - random size and number, rough volume fraction	Evaporation through holes - highly controllable - uniform size, defined number and position
Patternable material	Simple restriction: 1) water dispersibility	Difficult restrictions: 1) water dispersibility 2) inertness to chemical reaction 3) side-effect of by-product	Simple restriction: 1) water dispersibility Various functional materials (e.g. bacteria, quantum dot, graphene, photoresist, particles, and so on)
Processing time	Long time: 2 h for 2 cm ²	Long time: hours for complete chemical reaction and evaporation of enclosed liquid	Short time: ~ 3 min for any size of area
Reliability	Low	Middle (stochastic defect control)	High (not stochastic defect control)
Producible pattern	None	- Various pattern (unit element = a straight liquid film) - One-way pattern on a post array	- Various pattern (unit element = a straight liquid film or a threefold junction) - Multi-way pattern on a post array
Potential in material patterning applications	Low potential Main reason: 1) low controllability	Low potential Main reason: 1) limited available material, 2) closed-cell structure, 3) time-consuming process	Should be potential Main reason: 1) various available material, 2) open-cell structure, 3) fast process
Example of application	- Transparent electrode	- Transparent electrode	- Transparent and flexible electrode - Multi-functional device (integration of multiple functional material pattern, Fig 4) - Low-cost and high-throughput nanoscale direct writing (Fig. 5)

Point by Point Responses to the Reviewers Comments

We appreciate all the reviewers' comments and suggestions on our manuscript because they were definitely helpful for us to improve the overall quality of the revised manuscript. We provide our point-by-point responses below. In particular, we elaborately clarified both the scientific and technological advances of our study in this revision in comparison with other references appearing to be relevant to each other. Please refer to the separate detailed document titled “Common Response Material.pdf” that is provided along with this response letter for all three reviewers. We would be glad to respond to any further questions and comments that the reviewers may have.

Reviewer #1 (Remarks to the Author):

The manuscript reports on micro/nanofluidic approach for fabrication of 2D liquid foam arrays, which has a potential in patterning the materials in liquid at the micro- and nanoscales. In addition, the mechanism of 2D liquid foam generation with exclusion of Ostwald ripening was discussed. The manuscript is clearly written and the conclusions are adequately supported by the obtained data.

Answer: We thank the reviewer for the positive review comments on the manuscript and the technological potential of our scientifically novel physical system (i.e., open-cell liquid foam generation).

However, the manuscript is not suitable for publication in Nature Communications because of lack of novelty. There is a report in the same journal ("A general patterning approach by manipulating the evolution of two-dimensional liquid foams", Z. Huang et al., Nature Commun., 2017, DOI: 10.1038/ncomms14110), in which a strategy to manipulate the evolution of 2D liquid foams beyond Ostwald ripening is described and closely related results are obtained. Unfortunately, the authors did not refer to this publication. I expect that the manuscript can be suitable for publication in journals in the field of colloids and interfaces, e.g. Colloids and Surfaces A, J. Colloid and Interface Sci., Langmuir.

Answer: We do appreciate the critical comment on the literature reported by Huang et al., Nat. Commun. 2017 [Ref. 22]. We also apologize for not citing the reference. In fact, both this work and the reference might have been inspired by the same article titled “Transparent electrodes fabricated via the self-assembly of silver nanowires using a bubble template” (published by Tokuno et al., Langmuir 2012). This article had been properly cited in our original manuscript. It appears that our work has a close relation with these two references. However, we can emphasize many scientific and technological advances and unique differences of our work compared to the aforementioned references. First of all, we briefly summarize the distinguishing originality and novelty of our study as follows.

- 1. Complete exclusion of Ostwald ripening (this work) vs. modulation of Ostwald ripening (Ref. 22).** Our technique further advances the conventional closed-cell 2D liquid foam generation mechanism modulating Ostwald ripening by engineering natural 2D liquid foam, to a

scientifically unprecedented open-cell 2D liquid foam generation and control mechanism.

2. **Physical spontaneous evaporation (this work) vs. chemical reaction (Ref. 22).** Our technique for the first time describes a novel and practical generation and control mechanism of uniform open cell arrays to the best of our knowledge.
3. **Various shapes of liquid film networks as designed.** We demonstrate many liquid film networks and patterns as designed such as periodic threefold liquid film junctions that have never been produced in other literatures.
4. **Multi-way foam control (this work) vs. one-way foam control (Ref. 22).** Our technique makes it possible to repeat the same liquid patterning process over and over on a single substrate, enabling multiple integration and superposition of heterogenous material patterns on various substrates.
5. **Versatile and synergistic compatibility of our technique with conventional micro- and nanofabrication processes.** Our technique can be combined with conventional nanofabrication techniques so that only advantages of the two techniques can be taken at the same time.

→ Please refer to the separate document titled “Common_Response_Material.pdf” that is provided along with this response letter for more details of the novelty of our study.

That is, our study as well as Ref. 22 showed good control of liquid foam at micro- and nanoscale by reversing or eliminating Ostwald ripening. Both studies seem to have their own scientific originality in contrast. Again, our study employs a different microfluidic approach consisting of a hole array and a post array, and it is based on a physical foam generation mechanism relying only on a spontaneous evaporation (i.e., control of open-cell liquid foam). On the other hands, Ref. 22 is based on post array and a gas-generating chemical reaction (i.e., control of closed-cell liquid foam). To highlight the novelty and originality in the revised manuscript, we changed the title, added more references [Ref. 21, 22], conducted more experiments (Fig. 3j, k) even showing liquid patterning without any post array (line 218 on page 9; Fig. 3j). We also newly discuss the scientific advances of the open system (line 51 ~ 69 in introduction; line 304 ~ 317 in discussion). Moreover, we clarified that the open system can satisfy even more basic requirements in liquid-mediated patterning fields than other reported closed-cell liquid foams from the viewpoint of patternable material choices, reliability of large-area patterning, processing time, multiple heterogeneous material integration, and synergistic combination with common lithographic processes (line 318 ~ 340 in discussion). Therefore, we are certain that the open-cell foam generation and control technique will make a big breakthrough in liquid-mediated patterning of various materials. Of course, we acknowledge the reverse Ostwald ripening study by Huang et al. because it also should be scientifically novel. And they envisioned remarkable potential of foam control for nanopatterning.

We appreciate again that the comments given by the reviewer significantly helped to improve the scientific and technological novelty of our work as a potential article in Nat. Commun. journal. We would be glad to respond to any further questions and comments that you may have. Thank you very much.

Reviewer #2 (Remarks to the Author):

I enjoyed reading this manuscript. The authors used clever microfluidic designs to stabilize a monolayer bubble emulsion (foam) during evaporation: air vents to allow vapor escape, pillars to stabilize against Oswald ripening and geometric optimization to pin the lamellae. The result is a beautiful hexagonal lamella structure without topological defects, whose films are sub-micron in dimension. With colloids within the film, patterned colloidal aggregations result after evaporation, with micro-lattices and nanoscale sheets. I believe the paper is suitable for publication in Nature Communication, if these designs are new. I am not sufficiently familiar with the literature to assess if this is true.

Answer: We thank the reviewer for the positive review comments. At the same time, we are sorry for not giving enough discussion on the originality and/or novelty of our work. We should have been more considerate in the discussion for the diverse readership of Nat. Commun., although there are only a few relevant references regarding “2D foam generation and control” beyond Oswald ripening. In the revised manuscript, we elaborately emphasized many scientific and technological advances of our work compared to the references. The originality and novelty of our study are summarized shortly:

- 1. Complete exclusion of Oswald ripening (this work) vs. modulation of Oswald ripening (Ref. 22).** Our technique further advances the conventional closed-cell 2D liquid foam generation mechanism modulating Oswald ripening by engineering natural 2D liquid foam, to a scientifically unprecedented open-cell 2D liquid foam generation and control mechanism.
- 2. Physical spontaneous evaporation (this work) vs. chemical reaction (Ref. 22).** Our technique for the first time describes a novel and practical generation and control mechanism of uniform open cell arrays to the best of our knowledge.
- 3. Various shapes of liquid film networks as designed.** We demonstrate many liquid film networks and patterns as designed such as periodic threefold liquid film junctions that have never been produced in other literatures.
- 4. Multi-way foam control (this work) vs. one-way foam control (Ref. 22).** Our technique makes it possible to repeat the same liquid patterning process over and over on a single substrate, enabling multiple integration and superposition of heterogenous material patterns on various substrates.
- 5. Versatile and synergistic compatibility of our technique with conventional micro- and nanofabrication processes.** Our technique can be combined with conventional nanofabrication techniques so that only advantages of the two techniques can be taken at the same time.

→ Please refer to the separate document titled “Common_Response_Material.pdf” that is provided along with this response letter for more details of the novelty of our study.

That is, our study as well as Ref. 22 showed good control of liquid foam at micro- and nanoscale by reversing or eliminating Ostwald ripening. Both studies seem to have their own scientific originality in contrast. Again, our study employs a different microfluidic approach consisting of a hole array and a post array, and it is based on a physical foam generation mechanism relying only on a spontaneous evaporation (i.e., control of open-cell liquid foam). On the other hands, Ref. 22 is based on post array and a gas-generating chemical reaction (i.e., control of closed-cell liquid foam). To highlight the novelty and originality in the revised manuscript, we changed the title, added more references [Ref. 21, 22], conducted more experiments (Fig. 3j, k) even showing liquid patterning without any post array (line 218 on page 9; Fig. 3j). We also newly discuss the scientific advances of the open system (line 51 ~ 69 in introduction; line 304 ~ 317 in discussion). Moreover, we clarified that the open system can satisfy even more basic requirements in liquid-mediated patterning fields than other reported closed-cell liquid foams from the viewpoint of patternable material choices, reliability of large-area patterning, processing time, multiple heterogeneous material integration, and synergistic combination with common lithographic processes (line 318 ~ 340 in discussion). Therefore, we are certain that the open-cell foam generation and control technique will make a big breakthrough in liquid-mediated patterning of various materials. Of course, we acknowledge the reverse Ostwald ripening study by Huang et al. because it also should be scientifically novel. And they envisioned remarkable potential of foam control for nanopatterning.

We would be glad to respond to any further questions and comments that you may have. Thank you very much.

My only technical criticism is on the arguments of how smaller channel height can pin the meniscus by ensuring the bottom contact angle does not exceed the advancing contact angle. The specific advancing contact angle for their surface was not reported and it is not clear if it even exists for the hydrophilic substrate used. It is also not clear that the simple interfacial shape depicted is actually observed. Surely when the lamellae and plateau are formed, such a simplistic interface model is not accurate and the advancing contact angle argument breaks down, as the contact line is surely receding. This argument can be tightened more. A better argument is that the expanding bubble will make contact with the bottom substrate before the interface at the pinned location makes contact with the top substrate. This ensures that the meniscus curvature will always render the liquid pressure negative due to capillarity, thus suppressing any liquid pressure gradient that can prematurely drive meniscus motion.

Answer: We appreciate the reviewer for the careful assessment on our study. We are very sorry to make the reviewer confused because we made a mistake in the terminology. That is, the ‘advancing’ should have been the ‘receding’ in the manuscript. To resolve the reviewer’s concern about the liquid pinning, we conducted numerical simulation to further investigate the cause of the liquid pinning at the bottom side of the holes. And we corrected and strengthened our argument in the revised manuscript (line 172 ~ 185 on page 8).

First, the main reason for the liquid pinning can be explained by the fact that the decrement of h_p values results in the increase of capillary forces acting on the membrane, subsequently resulting in the

local collapse of the membrane between nearby posts (see illustration in Fig. 2e). This is well supported by the updated experimental result as shown in Supplementary Fig. 6c (indeed, this was suggested by the reviewer #3 as well). That is, the local membrane collapse makes uneven height, denoted as gaps, near the holes so that the interfaces at the smaller height/gap are pinned at the holes. This is also experimentally supported in Fig. 2e, in which distorted circular interfaces expand toward the nearest post from the holes. This is because the gaps near posts are greater than those in the middle of posts by the collapse of the membrane. As a result, the liquid-air interfaces propagate toward the nearest post of which the gap is also highest.

Second, we conducted numerical simulations to investigate the liquid pinning at the bottom side of the membrane holes during evaporation. Fig. R1a represents a computational domain with detail geometry information. The computational domain includes both the liquid (i.e., water) and gas (i.e., air) phase. To observe an evaporation process, we needed to solve three equations. First, Navier-Stokes equations with continuity were used to calculate the fluid velocity of both the liquid and the gas separately as follows:

$$\rho \left(\frac{\partial \mathbf{u}}{\partial t} + \mathbf{u} \cdot \nabla \mathbf{u} \right) = -\nabla p + \nabla \cdot \left(\mu (\nabla \mathbf{u} + (\nabla \mathbf{u})^T) - \frac{2}{3} \mu (\nabla \cdot \mathbf{u}) \mathbf{I} \right) \quad (1)$$

$$\rho \nabla \cdot \mathbf{u} = 0 \quad (2)$$

where ρ and μ denote the density and dynamic viscosity of the fluids (i.e., air and water), respectively. Second, Fick's law was used to calculate the diffusion and advection of water vapors:

$$\mathbf{J} = -D \nabla c + \mathbf{u} \cdot \nabla c \quad (3)$$

where \mathbf{J} represents the evaporation flux at the liquid-gas interface and c does the vapor concentration. Third, the general heat transfer equation in conjunction with Fick's law was used to calculate the evaporation flux:

$$\rho C_p \mathbf{u} \cdot \nabla T + \nabla \cdot \mathbf{q} = Q \quad (4)$$

where C_p , \mathbf{q} , and Q represent the heat capacity of the fluids, heat flux, and latent heat, respectively. The evaporation flux was calculated from the enthalpy difference of the liquid-water and the vapor with the following equation:

$$M_f = \frac{Q}{h_{fg}} \quad (5)$$

where M_f represents the evaporation flux in mass and h_{fg} does the enthalpy difference (i.e., $h_{fg} = h_g - h_f$). The detailed boundary conditions of all the walls are summarized in Table R1.

We conducted simulations for two different cases as shown in Fig. R1b. For this, we fixed the gap at the left-side edge of the hole to be 10 μm , while varying the gap at the right-side to be 8 and 9.8 μm . Therefore, the gap difference between the left-side and the right-side became $\Delta\text{gap} = 2$ and 0.2 μm , respectively. Then, the liquid-gas interface was tracked over time and the three contour plots at the initial, intermediate, and final state are shown, respectively. First of all, for $\Delta\text{gap} = 2$ μm , the liquid-gas interface at the left-side gradually receded leftward, while the concavity of liquid-air interface at the right-side

remained constant from the intermediate to the final state. That is, the interface at the right-side was pinned at the bottom side of the hole. Next, for $\Delta_{\text{gap}} = 0.2 \mu\text{m}$, from the initial to the intermediate state, the evaporation process and movement of the liquid-air interface seemed to be similar as the previous simulation result for $\Delta_{\text{gap}} = 2 \mu\text{m}$. However, after the intermediate state, not only the left-side interface but also the right-side receded without the pinning at the bottom side of hole, although the left meniscus receded slightly earlier and faster than the right one. Fig. R1c clearly describes the pinning of the liquid-air interface after arriving at the edge; three timelines of the liquid-gas interface at the right side of the hole are clearly different. Hence, if there is an influential gap difference, the receding of the liquid-air interfaces only occur at the larger gap not at the smaller gap. Such migration and pinning of the liquid-air interface have been experimentally observed and demonstrated by Kang et al. and can be well explained by using the Yong-Laplace equation and geometrical parameters [Ref. 31]. In sum, we believe that these simulation results not only adamantly support our experimental observation shown in Fig. 2e but also clarify the priority rule of the moving direction of the liquid-gas interface.

Fig. R1. Numerical simulation results obtained by using COMSOL Multiphysics. (a) 2D modelling domain shows a side view of the hole structure. (b) Contour plots showing time-sequential vapor concentration. (c) Tracking of the liquid-gas interface over time after arriving at the edge of the hole. The red, blue, and green lines represent timelines at t_1 , t_2 , and t_3 ($t_1 < t_2 < t_3$), respectively.

Table R1. Boundary conditions in which T_0 , c_0 , σ , P_{sat} , and R represent ambient temperature, related humidity, surface tension, saturated pressure, and gas constant of water, respectively.

Boundary number	1	2	3 - 4	5 - 9
Condition	Open	Liquid-air interface	Periodic	Wall
Navier-Stokes	Normal stress = 0	$\mathbf{n} \cdot \mathbf{F}_{\text{gas}} - \mathbf{n} \cdot \mathbf{F}_{\text{liquid}} = \sigma(\nabla \cdot \mathbf{n}) - \nabla \cdot \sigma$	$\mathbf{u}_{\text{source}} = \mathbf{u}_{\text{destination}}$	$\mathbf{u} = 0$
Fick's law	$C_0 = 40\%$	Equation of ideal gas state	No define	$-\mathbf{n} \cdot \mathbf{D} \nabla c = 0$
Heat transfer	$T_0 = 25^\circ\text{C}$	$\frac{P_{\text{sat}}}{R \cdot T}$	$\mathbf{T}_{\text{source}} = \mathbf{T}_{\text{destination}}$	$-\mathbf{n} \cdot \mathbf{q} = 0$

Reviewer #3 (Remarks to the Author):

The manuscript relates to a method of making complex micro/nanoscale patterns using controlled drying of 2D foams trapped between a structured microtextured surface, and a microporous surface. The alignment of micropores on the microposts allows for nearly uniform evaporation of the foam resulting in naturally thinned nano/micro-liquid patterns pinned along the microposts. Such liquid patterns can be used to make solid patterns by simply adding either nanoparticles or a water-soluble polymer into the evaporating solution. The authors have demonstrated that by cleverly playing between the geometrical features of the underlying micropost surface and the overlaid microporous surface, multi-patterned nano/micro-structures can be obtained. The manuscript is quite well-written, the authors have done quite exhausted study of different parameters that can influence the design of nano/micro-structures with the technique. To the reviewer, the manuscript appears to be suitable for the journal - provided the authors address the concerns mentioned below.

Answer: We thank the reviewer for the positive assessment on our study and manuscript.

To be clear, the idea of using foam evaporation guided by the micropost surfaces is not new. It has previously been demonstrated in the manuscript titled “A general patterning approach by manipulating the evolution of two-dimensional liquid foams”. It is unclear why the authors did not cite this work. This is unacceptable. Although the basic idea has been demonstrated before, the reviewer does appreciate the use of overlaid microporous surface to control the foam evaporation and how that provides as an extra-control over the design of the nano/micro-structures one can obtain using this method.

Answer: We thank the reviewer for recommending the reference reported by Huang et al. (Nat. Commun. 2017) and guiding us to revise our manuscript. Due to the sharp and comprehensive review comments, we compared our work with the reference and then could improve the overall quality of our work. We also apologize for not citing the article although our work has a close relation with it apparently. It seems to be because both our work and the article have been inspired by another reference entitled “Transparent electrodes fabricated via the self-assembly of silver nanowires using a bubble template” (reported by Tokuno et al., Langmuir 2012). Note that this reference was cited in our manuscript before.

For clarity, we emphasized many scientific and technological advances of our work compared to the references. The originality and novelty of our study are summarized shortly:

- 1. Complete exclusion of Ostwald ripening (this work) vs. modulation of Ostwald ripening (Ref. 22).** Our technique further advances the conventional closed-cell 2D liquid foam generation mechanism modulating Ostwald ripening by engineering natural 2D liquid foam, to a scientifically unprecedented open-cell 2D liquid foam generation and control mechanism.
- 2. Physical spontaneous evaporation (this work) vs. chemical reaction (Ref. 22).** Our technique for the first time describes a novel and practical generation and control mechanism of uniform

open cell arrays to the best of our knowledge.

3. **Various shapes of liquid film networks as designed.** We demonstrate many liquid film networks and patterns as designed such as periodic threefold liquid film junctions that have never been produced in other literatures.
4. **Multi-way foam control (this work) vs. one-way foam control (Ref. 22).** Our technique makes it possible to repeat the same liquid patterning process over and over on a single substrate, enabling multiple integration and superposition of heterogenous material patterns on various substrates.
5. **Versatile and synergistic compatibility of our technique with conventional micro- and nanofabrication processes.** Our technique can be combined with conventional nanofabrication techniques so that only advantages of the two techniques can be taken at the same time.

→ Please refer to the separate document titled “Common_Response_Material.pdf” that is provided along with this response letter for more details of the novelty of our study.

That is, our study as well as Ref. 22 showed good control of liquid foam at micro- and nanoscale by reversing or eliminating Ostwald ripening. Both studies seem to have their own scientific originality in contrast. Again, our study employs a different microfluidic approach consisting of a hole array and a post array, and it is based on a physical foam generation mechanism relying only on a spontaneous evaporation (i.e., control of open-cell liquid foam). On the other hands, Ref. 22 is based on post array and a gas-generating chemical reaction (i.e., control of closed-cell liquid foam). To highlight the novelty and originality in the revised manuscript, we changed the title, added more references [Ref. 21, 22], conducted more experiments (Fig. 3j, k) even showing liquid patterning without any post array (line 218 on page 9; Fig. 3j). We also newly discuss the scientific advances of the open system (line 51 ~ 69 in introduction; line 304 ~ 317 in discussion). Moreover, we clarified that the open system can satisfy even more basic requirements in liquid-mediated patterning fields than other reported closed-cell liquid foams from the viewpoint of patternable material choices, reliability of large-area patterning, processing time, multiple heterogeneous material integration, and synergistic combination with common lithographic processes (line 318 ~ 340 in discussion). Therefore, we are certain that the open-cell foam generation and control technique will make a big breakthrough in liquid-mediated patterning of various materials. Of course, we acknowledge the reverse Oswald ripening study by Huang et al. because it also should be scientifically novel. And they envisioned remarkable potential of foam control for nanopatterning.

We would be glad to respond to any further questions and comments that you may have. We appreciate the comment and suggestion again.

The authors appear to have done quite some extensive experiments varying the post radius, distance, hole radius etc. Depending upon such geometric parameters, different results are obtained. It would be very useful to the readers if the authors could provide a catalog of their experiments and the final image in the supplementary file.

Answer: It is such a good suggestion to provide a catalogue for potential readers and researchers who will need to utilize our technique. As suggested, we added a catalogue by reorganizing all the experimental conditions and corresponding results (see Supplementary Table. 1). The catalogue includes quantification of defect generation (refer to Fig. 2c), structural stability of the device including collapse (refer to Fig. 2d), multiple effects of the post height (refer to Fig. 2e, f), and designing examples of various liquid patterns through the combination of posts and holes (refer to Fig. 3). We hope the catalogue will be helpful to minimize extensive experiments. Thank you again for the good suggestion.

The authors should provide the images of the foam structures based on which Fig 2c and Fig 2g were obtained. Such images could be part of the suggested catalog as mentioned above – or these could be separate figures in the SI itself.

Answer: We added more images of experimental results in Fig. 2c and Fig. 2g (Supplementary Fig. 4 and Supplementary Fig. 7). We also added the experimental conditions of these results in the catalogue as well.

It is unclear whether Fig2c data was obtained by using samples with different r_p ? Authors should clarify that they sequentially varied the post diameter to obtain Fig 2c. The authors should also mention geometrical parameters (including h_p) of samples shown in the figure captions.

Answer: Data of Fig. 2c was obtained by varying the r_p values when $d = 130$ or $200 \mu\text{m}$ and $h_p = 25$ (line 147 on page 6). Wherever necessary in the manuscript, we carefully revised to clarify the key geometrical parameters of the experiments (line 144, 147, 167, 193, and so on; especially the suggested catalogue).

Authors need to provide more description of the alignment of the PUA over PDMS structures. Authors describe PDMS/NOA frames – but it is unclear how these frames were used? Were they attached to PUA membranes? The thickness of PUA membranes is really thin. How did the authors avoid the folding of the structures and overlaying it over the structures?

Answer: We added more detailed description about the alignment process and the adhesion between them (line 320 ~ 383 on page 16). In short, we manually aligned the PUA membrane (a microhole array) with the frame (a micropost array), observing them with an optical microscope by sliding the membrane in the presence of water as a lubricant. During the evaporation of the water, capillary forces allowed conformal contact between them and then the membrane was adhered to the frame by van der Waals force interaction. **[Redacted]**. In addition, we added an image and the mechanical property of the PUA membrane to help potential readers to identify the mechanical stability (Supplementary Fig.1c; line 369 on page 15). Although the thickness of the PUA membrane was about $25 \mu\text{m}$, its Young's modulus was large enough to handle without folding. From the added image,

we can see that the membrane is not susceptible to folding like a piece of common paper unless it is intentionally pressed.

Some of the materials information is essential and should be included in the main text itself. For example, the solution used, the molecular weight of PDMS, PUA etc are crucial information. Such information can be included in the main text or the figure captions.

Answer: We carefully revised the manuscript to include further detailed material information wherever necessary. For example, the conditions of used solutions (line 92 ~ 94 on page 4; line 271 on page 11) and the product number of the chemicals (line 351 and 357 on page 15). We newly added a reference about the composition of the PUA product [Ref. 40] of which quantitative composition is unfortunately unavailable due to the confidential intellectual property of the company (Minuta Technology, Korea). In case of PDMS and NOA 64 products, the well-known world-wide manufacturers are informed.

Minor point: Instead of saying “under surface tension when d is long”, it is better to say “under surface tension when d is large”. Similarly, instead of using word ‘Short’ – use ‘small’.

Answer: Thank you for the correction. As suggested, we changed “when d is long” to “when d is large” and “resulting shorter gaps” to “resulting smaller gap”.

Fig2d: The collapse of the overlaid membrane should also depend upon r_p and h_p . If r_p is large, then the membrane may not collapse. The reviewer imagines that there should be some d/r_p above which the collapse may occur. Can the authors provide a criterion on when the films would collapse as a function of geometrical parameters and the bending modulus of the overhang film?

Answer: We thank the reviewer for the good insight into the collapse mechanism of the membrane. It is obvious that the collapse mechanism of the membrane can be explained by such parameters. For clarity, we conducted an experiment to investigate the effect of h_p on the membrane collapse and the result is shown in Supplementary Fig. 6a, b, c, and g. In addition, we newly discussed the effects of the geometrical parameters (e.g., h_p , r_p) and mechanical properties of materials on the membrane collapse mechanism (line 169 ~ 171 on page 7). However, it is unfortunate that the additional quantitative characterization and/or modelling of the membrane collapse mechanism seems to be beyond the scope of this work.

Fig 2e: The effect of h_p has been shown for a single experiment. Is it universal behavior? For all samples with $d < 400 \mu\text{m}$, do you obtain distorted foams for $h_p < 10 \mu\text{m}$? If not, the authors should clarify/correct/edit the statement.

Answer: It is obvious that the reviewer completely understood our experiment and then gave us a very sharp question about it. In the revised manuscript (line 172 ~ 185 on page 8), we clarified that exact

universal effect of decreased h_p is increment of capillary force acting on the membrane. In addition, we clarified that we didn't obtain distorted foams for all sample with $h_p = 10 \mu\text{m}$. Meanwhile, we corrected the geometrical condition with equality signs rather than inequality signs for clarity. The new correction is based on new experiments (Supplementary Fig. 6a, c) showing results of condition at two more different d when $h_p = 10 \mu\text{m}$ in addition to the result shown in Fig 2e. For $d = 50 \mu\text{m}$ (e.g., $r_p = 30 \mu\text{m}$, $d = 50 \mu\text{m}$, and $h_p = 10 \mu\text{m}$), although the pinning-induced distortion of the liquid-air interface occurs, the experiment finally results in a well-ordered liquid pattern unlike the condition for $r_p = 30 \mu\text{m}$, $d = 100 \mu\text{m}$, and $h_p = 10 \mu\text{m}$. We guess that the ordered (Supplementary Fig. 6a) or not ordered (Fig. 2e) pattern resulted according to geometry-dependent elapsed degree of progress in the pinning-induced distortion. For $d = 200 \mu\text{m}$ (e.g., $r_p = 30 \mu\text{m}$, $d = 200 \mu\text{m}$, and $h_p = 10 \mu\text{m}$), a similar result as Fig. 2d occurs without the pinning-induced distortion. In other words, as h_p decreases capillary forces acting on the membrane increase, which is a well-known physical relation as reported in the reference [Ref. 32]. For example, a local collapse result is shown in Fig. 2e and Supplementary Fig. 6a whereas a global collapse result is shown in Supplementary Fig. 6c.

→ Fig. R1 shows our numerical simulation results on the pinning mechanism. Please refer to our answer to the question raised by Reviewer #2 for more details.

Fig. R1. Numerical simulation results obtained by using COMSOL Multiphysics. (a) 2D modelling domain shows a side view of the hole structure. (b) Contour plots showing time-sequential vapor concentration. (c) Tracking of the liquid-gas interface over time after arriving at the edge of the hole. The red, blue, and green lines represent timelines at t_1 , t_2 , and t_3 ($t_1 < t_2 < t_3$), respectively.

What happens if h_p is very large? Can authors provide some clarification on its possible effect?

Answer: In order to answer the question, we newly conducted experiments with $h_p = 100 \mu\text{m}$, which is 4-times greater than the normal experimental condition for the MNLP (Fig. 2f; Supplementary Fig. 6h, i; line 185 ~ 193 on page 8). For this condition, the menisci in the holes do not recede while only a penetrated liquid-air interface (i.e., meniscus) in the gap between the top surface of the frame and the bottom surface of the membrane keeps receding (refer to the conceptual illustration in Fig. 2f and Supplementary Fig. 6h). This is attributed to the fact that the liquid-air interfaces in a larger gap recede while those in a smaller gap are pinned according to the Young-Laplace equation. This is experimentally and theoretically well known in the literature [Ref. 31]. The reason for this result is fundamentally the same as the explanation for the result shown in the Fig 2e, which is mentioned in our answer right above. This is also supported by the result that the undesired expansion of the liquid-air interfaces does not occur when the diameter of the hole (i.e., $150 \mu\text{m} = 2 \times r_h$) is larger than $h_p = 100 \mu\text{m}$ (Supplementary Fig. 6i). Thank you for your comment.

Minor: The authors state “were sintered at 90 °C for 2 mins”. Was sintering done after each experiment?

Answer: The sintering process was performed after each patterning process for the multiple heterogeneous material patterns (refer to the line 210 on page 9).

The reviewer is very curious to know why the Fig.4 experiments worked. Why did the first deposition not alter the evaporative dynamics of foam. In other words, it appears that the patterns made no difference in how the foams evaporated. Why? Was it because they are too small in size? Was there no pinning on these patterns?

Answer: Thank you for your interest in Fig. 4. When producing the result in Fig. 4, we never observed noticeable disturbance of the evaporative dynamics of sub-micron foam structures. We also agree with the reviewer that the patterned surface area seems to be too small to affect the whole liquid-patterning process. In addition, to clearly answer your question about ‘influence of the first material deposition on the second liquid patterning process’, we obtained additional experimental results using a larger size of ridge structures (i.e., the first material deposition, Supplementary Fig. 8); see also the time-lapse movie of the second patterning process (Supplementary Movie 3). As the reviewer guessed, the receding of the liquid-air interfaces was disturbed by the large prepatterned ridge structures (middle image in Supplementary Fig. 8a). However, eventually the well-ordered second liquid-pattern was produced even though the height of the first ridge structure was as high as $6 \mu\text{m}$ (~25 % of h_p). This might be attributed to the mutual interaction among the liquid-air interfaces in a symmetrical manner. Interestingly, the first material pattern affects the deposition behaviour of the second materials (Supplementary Fig. 8b). For example, the first material pattern was well-covered with a monolayer of the second materials, resulting in an integrated micron-scale heterogeneous material structures in our platform. Thank you for your sharp question.

REVIEWERS' COMMENTS:

Reviewer #1 (Remarks to the Author):

As indicated in the previous review, there is a report in the same journal ("A general patterning approach by manipulating the evolution of two-dimensional liquid foams", Z. Huang et al., Nature Comm., 2017, DOI: 10.1038/ncomms14110), in which a strategy to manipulate the evolution of 2D liquid foams beyond Ostwald ripening is described and similar data are obtained. Therefore, based on the lack of novelty, the recommendation was to reject the manuscript as not suitable for publication in Nature Communications.

In response to this decision, the authors consider certain advantages of their work as compared with the cited above (complete exclusion of Ostwald ripening, generation and control mechanism of uniform open cell arrays, design of various shapes of liquid film networks, multi-way foam control, etc.).

The authors changed the title, added more references and performed additional experiments, which really succeeded in strengthening the manuscript. Nevertheless, I insist on insufficiency of novelty compared to previously reported data to consider the manuscript as suitable for publication in Nature Communications.

Reviewer #2 (Remarks to the Author):

The authors have responded to the two major criticisms from the three reviews:

1. Comparison to a key prior work;
2. The mechanism behind the suppression of Oswald ripening by contact line pinning.

I believe the authors have shown their work is sufficiently new compared to earlier work. Their understanding and explanation of the contact-line dynamics remain dogmatic and excessively reliant on Comsol simulation, although they did correct a key misrepresentation of the contact line dynamics—it recedes instead of advances.

While a more incisive explanation of the key contact line dynamics would have elevated the scientific level of the paper, I believe the current version does exceed the bar for a short Nature Comm report, as it contains detailed study of a new multiphase microfluidic design that produces wonderful dry foam "crystals" that may have translational implications.

I hence recommend publication of the revised version.

Reviewer #3 (Remarks to the Author):

The authors have done a very good job in answering this reviewers questions, and made sufficient modifications in the manuscript based on the suggestions. The reviewer is pleased with their response and their efforts. I do not have any further questions/comments, and I am pleased to recommend this manuscript for the journal.

Response to the reviewers' comments

Reviewer #1 (Remarks to the Author):

As indicated in the previous review, there is a report in the same journal ("A general patterning approach by manipulating the evolution of two-dimensional liquid foams", Z. Huang et al., Nature Comm., 2017, DOI: 10.1038/ncomms14110), in which a strategy to manipulate the evolution of 2D liquid foams beyond Ostwald ripening is described and similar data are obtained. Therefore, based on the lack of novelty, the recommendation was to reject the manuscript as not suitable for publication in Nature Communications. In response to this decision, the authors consider certain advantages of their work as compared with the cited above (complete exclusion of Ostwald ripening, generation and control mechanism of uniform open cell arrays, design of various shapes of liquid film networks, multi-way foam control, etc.). The authors changed the title, added more references and performed additional experiments, which really succeeded in strengthening the manuscript. Nevertheless, I insist on insufficiency of novelty compared to previously reported data to consider the manuscript as suitable for publication in Nature Communications.

Answer: We appreciate positive assessments on improvements of the revisions. Though we understand the reviewer is careful about the novelty issue, we are certain that we tried our best to clarify both the scientific and technological advances of our work in the revised manuscript compared to the previous report. Thank the reviewer again for leading us to think novelty of our work in depth during the revisions.

Reviewer #2 (Remarks to the Author):

The authors have responded to the two major criticisms from the three reviews: 1. Comparison to a key prior work; 2. The mechanism behind the suppression of Oswald ripening by contact line pinning.

I believe the authors have shown their work is sufficiently new compared to earlier work. Their understanding and explanation of the contact-line dynamics remain dogmatic and excessively reliant on Comsol simulation, although they did correct a key misrepresentation of the contact line dynamics—it recedes instead of advances. While a more incisive explanation of the key contact line dynamics would have elevated the scientific level of the paper, I believe the current version does exceed the bar for a short Nature Comm report, as it contains detailed study of a new multiphase microfluidic design that produces wonderful dry foam “crystals” that may have translational implications. I hence recommend publication of the revised version.

Answer: We do appreciate the final recommendation. And we thank the reviewer for the comment that further theoretical study on contact line dynamics would show better scientific level of the new microfluidic design and thank for understanding that the further study would exceed scope of the manuscript. We believe that the future study on the dynamics will be good topic to us for follow-up study of the manuscript.

Reviewer #3 (Remarks to the Author):

The authors have done a very good job in answering this reviewers questions, and made sufficient modifications in the manuscript based on the suggestions. The reviewer is pleased with their response and their efforts. I do not have any further questions/comments, and I am pleased to recommend this manuscript for the journal.

Answer: We do appreciate all your comments and the final recommendation.